# Quantum robustness of the toric code in a parallel field on the honeycomb and triangular lattice

Viktor Kott, Matthias Mühlhauser, Jan Alexander Koziol, Kai Phillip Schmidt[⋆]

Department of Physics, Staudtstraße 7, Friedrich-Alexander-Universität Erlangen-Nürnberg, Germany

⋆ kai.phillip.schmidt@fau.de

June 11, 2024

## Abstract

We investigate the quantum robustness of the topological order in the toric code on the honeycomb lattice in the presence of a uniform parallel field. For a field in $z$-direction, the low-energy physics is in the flux-free sector and can be mapped to the transverse-field Ising model on the honeycomb lattice. One finds a second-order quantum phase transition in the 3D Ising⋆ universality class for both signs of the field. The same is true for a postive field in $x$-direction where an analogue mapping in the charge-free sector yields a ferromagnetic transverse-field Ising model on the triangular lattice and the phase transition is still 3D Ising⋆. In contrast, for negative $x$-field, the charge-free sector is mapped to the highly frustrated antiferromagnetic transverse-field Ising model on the triangular lattice which is known to host a quantum phase transition in the 3D XY⋆ universality class. Further, the charge-free sector does not always contain the low-energy physics for negative $x$-fields and a first-order phase transition to the polarized phase in the charge-full sector takes place at larger negative field values. We quantify the location of this transition by comparing quantum Monte Carlo simulations and high-field series expansions. The full extension of the topological phase in the presence of $x$- and $z$-fields is determined by perturbative linked-cluster expansions using a full graph decomposition. Extrapolating the high-order series of the charge and the flux gap allows to estimate critical exponents of the gap closing. This analysis indicates that the topological order breaks down by critical lines of 3D Ising⋆ and 3D XY⋆ type with interesting potential multi-critical crossing points. All findings for the toric code on the honeycomb lattice can be transferred exactly to the toric code on the triangular lattice.

# 1   Introduction

Long-range entangled topologically ordered quantum phases [1–3] are actively investigated over the last decades due to their relevance for the fractional quantum Hall effect [4, 5], for frustrated quantum magnets [6, 7] as well as for simulation of strongly correlated systems in quantum-optical platforms [8–12]. Such phases have exotic properties like elementary anyonic excitations with fractional statistics [13, 14]. At the same time there is the fascinating concept to build a topological quantum computer which is protected from local decoherence [15, 16].

One fundamental aspect in the research about topological phases is to study their quantum robustness in the presence of perturbations and to understand quantum phase transitions out of such phases. Contrary to conventional symmetry broken phases, no local order parameters exist and Landau-Ginzburg theory does not apply. As a consequence, an extended theoretical framework needs to be developed which is capable to describe such phase transitions. One possibility is in terms of condensing bosonic quasiparticles which is dubbed topological symmetry breaking [17–21]. This has been verified in microscopic models for phase transitions between topological and non-topological phases in various cases [22–38].

Kitaev's toric code [15] is an exactly solvable two-dimensional quantum spin model, which displays a topologically ordered ground state. It has been used as an ideal starting point to understand many fundamental aspects of topological quantum phases. This includes the quantum robustness and the associated topological phase transitions of the conventional toric code on the square lattice in the presence of a uniform field [22–26, 28–30, 33, 35, 36, 38–40]. The quantum phase diagram of the toric code in a field is very rich displaying planes of first- and second-order quantum phase transition [22–26, 28–30, 33, 35, 36, 38–40] as well as interesting multi-critical lines [25, 28, 30, 38]. Second-order phase transitions are generically in the 3D Ising* universality class between the topological phase at small fields and the symmetry unbroken polarized phase at large fields where the spins align along the direction of the magnetic field [22–25, 28–30, 33, 35, 36, 38–40]. For single parallel fields, this behaviour can be fully understood by the well-known duality between $\mathbb{Z}_2$ gauge theories and unfrustrated transverse field Ising models (TFIMs) in the charge-free or flux-free sector [41]. The continuous quantum phase transition out of the topological phase then corresponds to the condensation of charges or fluxes, which are defined on the (dual) square lattice.

For the toric code in a field on the square lattice the physical properties of charge and flux quasi-particles are identical when interchanging $x$- and $z$-fields because both quasi-particle types live on a (dual) square lattice. This symmetry leads to a perfectly symmetric quantum

phase diagram in the $xz$-plane. Furthermore, the phase diagram is independent of the sign of the parallel fields due to the bipartite structure and self-duality of the square lattice. It is therefore interesting to investigate how the quantum phase diagram is altered if this symmetry is broken and how the quantum criticality is changed when the charge or flux degrees of freedom are located on non-bipartite lattices and geometric frustration can be present. In Ref. [31] duality mappings for single parallel fields to fully frustrated Ising models have already been established and exciting behaviour has been demonstrated, e.g., an enhanced stability of the topological phase for the perturbed toric code on the dice lattice.

In this work, we investigate both questions for the toric code on the honeycomb lattice using high-order linked-cluster expansions as well as duality transformations and quantum Monte Carlo simulations for single parallel fields. In contrast to the square lattice, the quantum phase diagram is asymmetric in the $xz$-field plane because elementary charge and flux excitations behave fundamentally different. Indeed, their dynamics takes place on different lattice types, namely the honeycomb lattice for charges and its dual triangular lattice for fluxes. Our analysis reveals that the topological order breaks down by critical lines of 3D Ising$^\star$ [40] and 3D XY$^\star$ type [42–45] with interesting potential multi-critical crossing points. For single parallel fields the quantum critical behaviour can be quantitatively understood by duality mappings transverse-field Ising models (TFIM) and the absence or presence of geometric frustration. We further observe that all findings for the toric code on the honeycomb lattice can be transferred exactly to the toric code on the triangular lattice.

The paper is organized as follows. In Sec. 2 we introduce the toric code in a field on the honeycomb lattice. In Sec. 3 we present all relevant aspects of the applied methods. Results for single-field cases are discussed in Sec. 4, while results for the quantum robustness of the topological phase in the $xz$-plane are given in Sec. 5. Final conclusions are drawn in Sec. 6.

## 2 Honeycomb toric code in a field

The Hamiltonian of the toric code on the honeycomb lattice in a uniform field reads

$$
\begin{aligned}
H_{\circ}^{\text{tcf}} &= -\frac{1}{2}\sum_{\curlywedge} X_{\curlywedge} - \frac{1}{2}\sum_{\circ} Z_{\circ} - \sum_i \vec{h}\cdot\vec{\sigma}_i, \\
&= H_{\circ}^{\text{tc}} - \sum_i \vec{h}\cdot\vec{\sigma}_i,
\end{aligned}
\tag{1}
$$

where $\vec{\sigma}_i = (\sigma_i^x, \sigma_i^y, \sigma_i^z)$ is the Pauli vector acting on spin $i$ located on the edge of the honeycomb lattice and $\vec{h} = (h_x, h_y, h_z)$ is a uniform magnetic field acting on every spin (see Fig. 1). In this work, we focus on a field parallel to the plane defined by the direction of the Pauli matrices of the star and plaquette operators $\vec{h} = (h_x, 0, h_z)$. The star operators $X_{\curlywedge}$ and the plaquette operators $Z_{\circ}$ are both defined as products of Pauli matrices

$$
X_{\curlywedge} = \prod_{i\in\curlywedge} \sigma_i^x, \qquad\qquad Z_{\circ} = \prod_{i\in\circ} \sigma_i^z.
\tag{2}
$$

In contrast to the toric code on the square lattice, where star and plaquette operators are four-spin interactions, here the star operator $X_{\curlywedge}$ contains the Pauli operators on the three sites next to a vertex while the plaquette operator $Z_{\circ}$ contains six Pauli operators at the edges of a plaquette as illustrated in Fig. 1.

### 2.1 Toric code on the honeycomb lattice

The essential properties of the bare toric code on the honeycomb lattice are the same as its well known counterpart on the square lattice [15]. It is exactly solvable and displays topological

order, i.e., the ground state is highly entangled, the ground-state degeneracy scales with the genus $g$ of the system, and the elementary excitations are mutual Abelian anyons. In the following we describe some of these aspects in more detail.

All star and plaquette operators commute with each other:

$$[X_\lambda, X_{\lambda'}] = 0 \quad \forall \lambda, \lambda', \qquad [Z_\circ, Z_{\circ'}] = 0 \quad \forall \circ, \circ', \qquad [X_\lambda, Z_\circ] = 0 \quad \forall \lambda, \circ. \qquad (3)$$

Therefore, their eigenvalues $x_\lambda = \pm 1$ and $z_\circ = \pm 1$ are conserved quantities.

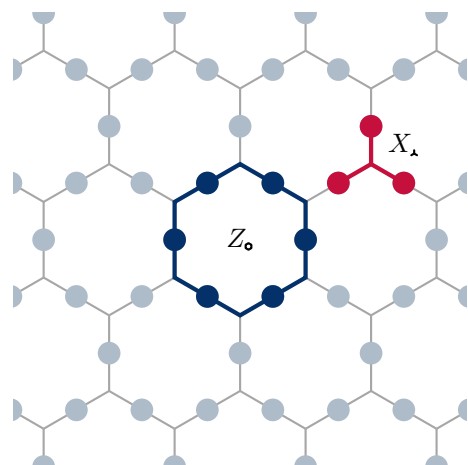

Figure 1: Illustration of the star (red) and plaquette (blue) operators on the honeycomb lattice. Filled circles denote the spin sites located at the edges of the lattice.

Ground states of the toric code have $x_\lambda = z_\circ = +1$ for all stars $\lambda$ and plaquettes $\circ$ and the ground-state energy is given by $E_0^{\mathrm{tc}} = -N_s/2 + N_p/2$ with $N_s$ ($N_p$) the number of stars (plaquettes). On an infinite open plane, the ground state is unique and can be constructed by projection operators as follows

$$|\mathrm{GS}\rangle = \mathcal{N} \prod_\lambda \frac{(1 + X_\lambda)}{2} \prod_\circ \frac{(1 + Z_\circ)}{2} |\mathrm{ref}\rangle, \qquad (4)$$

where $\mathcal{N}$ is a normalization factor and $|\mathrm{ref}\rangle$ an appropriate reference state which is often chosen as a fully polarized state in $x$- or $z$-direction, but can be any state with a finite overlap with $|\mathrm{GS}\rangle$. Products of operators $X_\lambda$ and $Z_\circ$ act trivially on $|\mathrm{GS}\rangle$. Furthermore, any contractible loop of $\sigma^z$ matrices on the honeycomb lattice or $\sigma^x$ matrices on its dual triangular lattice corresponds to the product of operators $X_\lambda$ or $Z_\circ$ contained in the loop, respectively [15].

The ground-state degeneracy depends on the genus $g$ of the system. For a torus with $g = 1$ not all star and plaquette operators are independent as

$$\prod_\lambda X_\lambda = \prod_\circ Z_\circ = \mathbb{1}. \qquad (5)$$

Accordingly, the number of spins in the system exceeds the number of independent star and plaquette operators by two. The remaining $\mathbb{Z}_2$ degrees of freedom can be attributed to eigenvalues $\pm 1$ of *non-local, non-contractible* loop operators. These loop operators are defined on closed paths on the (dual) lattice which wind around the torus in independent directions (see Fig. 2). They are defined as

$$X_{\tilde{\alpha}} = \prod_{i \in \tilde{\alpha}} \sigma_i^x, \qquad Z_\alpha = \prod_{i \in \alpha} \sigma_i^z \qquad (6)$$

with $X_{\tilde{\alpha}}^2 = Z_{\alpha}^2 = \mathbb{1}$. The indices $\alpha \in \{\tau, \pi\}$ denote the direction of the loop on the torus, either in toroidal direction $\tau$ or poloidal direction $\pi$ whereas a bar indicates a loop on the dual lattice. While all four types of these non-contractible loop operators commute with the Hamiltonian, they do not necessarily commute with each other

$$\left[X_{\tilde{\alpha}}, X_{\tilde{\beta}}\right] = 0, \tag{7}$$

$$\left[Z_{\alpha}, Z_{\beta}\right] = 0, \tag{8}$$

$$\left[X_{\tilde{\alpha}}, Z_{\beta}\right] = 2X_{\tilde{\alpha}} Z_{\beta} \left(1 - \delta_{\alpha,\beta}\right). \tag{9}$$

The first two commutation relations are trivial. The last relation arises form the fact, that loop operators $X_{\tilde{\alpha}}$ and $Z_{\beta}$ either intersect an odd number of times for different loop directions or an even number of times for the same direction. Furthermore, loop operators of the same type can be deformed into each other using products of operators $X_{\lambda}$ and $Z_{\mathbf{o}}$, respectively. So in order to enhance the set of operators $Z_{\mathbf{o}}$ and $X_{\lambda}$ to a complete set of commuting observables, it suffices to add two non-contractible loop operators which agree either in their direction ($\tau$ or $\pi$) or in the flavor of the contained Pauli matrices ($\sigma^x$ or $\sigma^z$). As the two $\mathbb{Z}_2$ degrees of freedom associated with these two non-contractible loop operators do not affect the energy, the ground-state degeneracy of the toric code on the torus is four. Generally, the ground-state degeneracy is $4^g$ as for the conventional toric code on the square lattice [15].

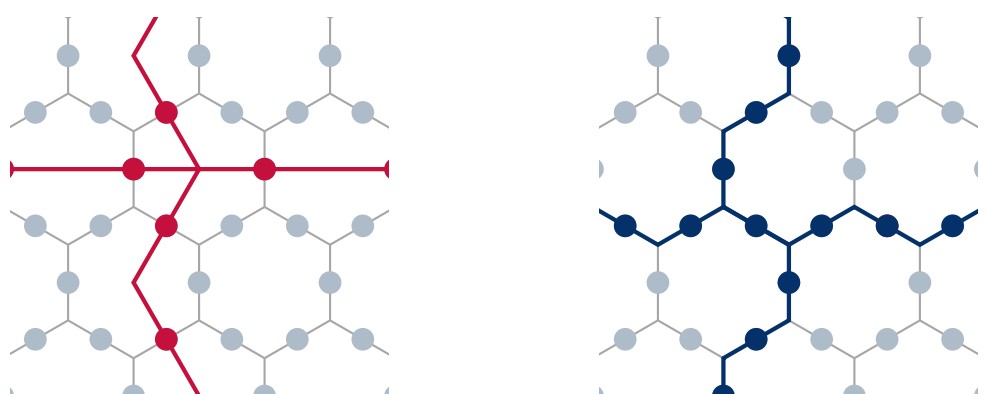

Figure 2: *Left*: $Z_{\tau}$ and $Z_{\pi}$ on a plane with periodic boundary conditions. *Right*: $X_{\tau}$ and $X_{\pi}$ on a plane with periodic boundary conditions.

Elementary excitations of the toric code correspond to negative eigenvalues of star and plaquette operators. Excitations $x_{\lambda} = -1$ are called charges and $z_{\mathbf{o}} = -1$ are called fluxes. These excitations can be created adjacent to a site $i$ by acting with $\sigma_i^z$ and $\sigma_i^x$ on a ground state. These topological excitations, which are situated at the vertices and plaquettes of the lattice, exhibit behaviour of hardcore bosons with a mutual Abelian anyonic statistics [15,30,46]. The string operators

$$S_z = \prod_{i \in p} \sigma_i^z \qquad \text{and} \qquad S_x = \prod_{i \in \bar{p}} \sigma_i^x, \tag{10}$$

applied to the ground state, create two excitations at the ends of the respective open path $p$ ($\bar{p}$) on the honeycomb (or its dual triangular) lattice [15].

A single charge or flux can be excited on an open plane by creating a pair of excitations and moving one of them to infinity. Using analogous definitions to Ref. [47], a single charge $x_{\lambda} = -1$ (flux $z_{\mathbf{o}} = -1$) can be represented by the one-charge state $|\vec{r}, l; \lambda\rangle$ (one-flux state $|\vec{r}; \mathbf{O}\rangle$):

$$|\vec{r}, l; \lambda\rangle = \prod_{i \in p_{\vec{r},l}} \sigma_i^z |\text{GS}\rangle, \qquad\qquad |\vec{r}; \mathbf{O}\rangle = \prod_{i \in \bar{p}_{\vec{r}}} \sigma_i^x |\text{GS}\rangle. \tag{11}$$

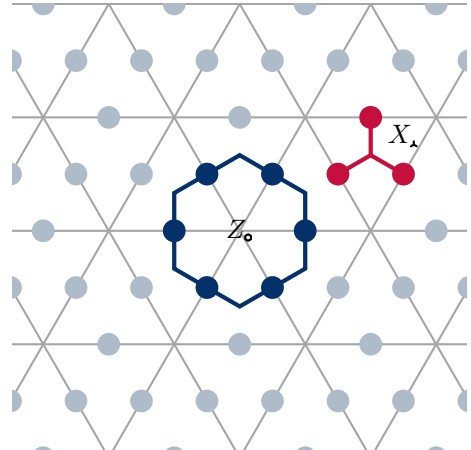 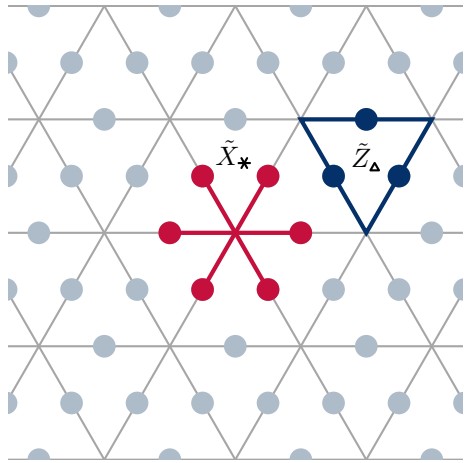

Figure 3: *Left*: The stabilizer operators $Z_{\circ}$, $X_{\curlywedge}$ of the toric code on the honeycomb lattice are on the dual triangular lattice. Compare to Fig. 1 to see that the spin degrees of freedom remain the same as on the original lattice. *Right*: Reinterpreting the original vertex operators $X_{\curlywedge}$ as plaquette operators $X_{\triangle}$ on the dual lattice and the orignal plaquette operators $Z_{\circ}$ as vertex operators $Z_{\maltese}$ on the dual lattice together with a rotated basis with $\tilde{\sigma}^z = \sigma^x$ and $\tilde{\sigma}^x = \sigma^z$ results in the stabilizers $\tilde{X}_{\maltese}$ and $\tilde{Z}_{\triangle}$ of the toric code on the triangular lattice.

Here $\vec{r}$ denotes the position of the unit cell, while $l \in \{1, 2\}$ denotes the position inside the unit cell of the honeycomb lattice. As in [47], $p_{\vec{r},l}$ ($\bar{p}_{\vec{r}}$) is a straight open path on the (dual lattice of the) honeycomb lattice which extends from the excitation to infinity in negative $x$-direction.

## 2.2 Mapping to the toric code on the triangular lattice

While in all the other parts of this paper we investigate the toric code on the honeycomb lattice, in this subsection we show how the results can be transferred exactly to the toric code on the triangular lattice by a duality mapping and a rotation. Indeed, the triangular lattice is dual to the honeycomb lattice and the very same spin sites reside on the links of both lattices. This fact becomes obvious comparing Fig. 1 to the left part of Fig. 3.

It is the nature of a duality mapping to identify the vertices of the original lattice with the tiles of the dual lattice and vice versa, so it is natural that six-spin plaquette $Z_{\circ}$ operators correspond to six-spin vertex operators $Z_{\maltese}$ and three-spin vertex operators $X_{\curlywedge}$ to three-spin plaquette operators $X_{\triangle}$. With this mapping to the dual lattice the Hamiltonian (1) becomes

$$H_{\text{dual}}^{\text{tcf}} = -\frac{1}{2}\sum_{\triangle} X_{\triangle} - \frac{1}{2}\sum_{\maltese} Z_{\maltese} - \sum_i \vec{h} \cdot \vec{\sigma}_i \,. \tag{12}$$

In a next step we express the Pauli matrices in a rotated basis. To this end we rotate the vector $\vec{\sigma} = (\sigma^x, \sigma^y, \sigma^z)^T$ by $\pi/2$ around the $y$-axis and then by $\pi$ around the $x$-axis

$$R_x(\pi)R_y(\pi/2) \cdot \vec{\sigma} = R_x(\pi) \cdot (\sigma^z, \sigma^y, -\sigma^x) = (\sigma^z, -\sigma^y, \sigma^x) = \tilde{\vec{\sigma}} \,. \tag{13}$$

Accordingly, the operators $X_{\triangle}$ and $Z_{\maltese}$ in the new basis read

$$\tilde{Z}_{\triangle} = \prod_{i \in \triangle} \tilde{\sigma}_i^z = \prod_{i \in \triangle} \sigma_i^x = X_{\triangle} \hookleftarrow X_{\curlywedge} \,, \tag{14}$$

$$\tilde{X}_{\maltese} = \prod_{i \in \maltese} \tilde{\sigma}_i^x = \prod_{i \in \maltese} \sigma_i^z = Z_{\maltese} \hookleftarrow Z_{\circ} \,, \tag{15}$$

which are the stabilizer operators of the toric code on the triangular lattice as illustrated in Fig. 3. Applying the same basis transformation to $\vec{h}$ we obtain $\tilde{\vec{h}} = (h_z, -h_y, h_x)$ and the Hamiltonian in the new basis reads

$$H_{\triangle}^{\text{tcf}} = -\frac{1}{2}\sum_{\triangle}\tilde{Z}_{\triangle} - \frac{1}{2}\sum_{\maltese}\tilde{X}_{\maltese} - \sum_i \tilde{\vec{h}} \cdot \tilde{\vec{\sigma}}_i, \tag{16}$$

which is the Hamiltonian of the toric code in a uniform magnetic field on the triangular lattice. As a consequence, the results for the toric code on the honeycomb lattice in a uniform field $\vec{h}$ are equivalently valid for the toric code on the triangular lattice in a uniform field $\tilde{\vec{h}}$. So, although in the other parts of this work we investigate the toric code on the honeycomb lattice, the results can be transferred exactly to the toric code in a field on the triangular lattice.

## 3 Methods

This section contains all relevant aspects of the applied methods which are needed for the discussion of the results presented in the following sections.

### 3.1 Low-field expansion

To locate the breakdown of the topological phase in a general $xz$-field, we use the pCUT method [48,49] to calculate the charge gap $\Delta^{\blacktriangle}$ and the flux gap $\Delta^{\circ}$ in the thermodynamic limit as a high-order series about the low-field limit. The analysis of the gap closing by extrapolation techniques (see Subsec. 3.2) allows to locate quantum critical points and determine associated critical exponents for a given field direction.

In the low-field expansion, we can express the field term $\sum_i h_x \sigma_i^x$ as $T_{+2}^{\text{f}} + T_{-2}^{\text{f}} + T_0^{\text{f}}$, since a field in x-direction either creates a flux-pair $T_{+2}^{\text{f}}$, annihilates a flux-pair $T_{-2}^{\text{f}}$ or moves a single flux by $T_0^{\text{f}}$ on the respective bonds. Similarly, the field in z-direction $\sum_i h_z \sigma_i^z$ can be expressed as $T_{+2}^{\text{c}} + T_{-2}^{\text{c}} + T_0^{\text{c}}$, as it creates a charge-pair, annihilates a charge-pair or moves a charge on the respective bonds. In both cases, $T_n = T_n^{\text{f}} + T_n^{\text{c}}$ fullfills the commutation relation $[H_{\circ}^{\text{tc}}, T_n] = nT_n$, with $n$ representing the net change in total charge and flux particle numbers resulting from the action of $T_n$. Hence, for finite fields, the system poses a formidable quantum many-body problem where the number of bare charges and fluxes is no longer conserved and elementary charge and flux excitations acquire a finite dispersion and interact. Using properties of the unperturbed toric code alongside this decomposition facilitates the application of the pCUT method [48, 49], which has been successfully applied to the conventional toric code on the square lattice subjected to a magnetic field [25, 30, 50].

Using the pCUT method, one can map the toric code in a field (1) to an effective Hamiltonian $H_{\text{eff}}$ which conserves quasi-particle numbers. The mapping is perturbatively exact up to the calculated order in the expansion parameters $h_x$ and $h_z$. This Hamiltonian satisfies $[H^{\text{tc}}, H_{\text{eff}}] = 0$, rendering it block-diagonal, thus reducing the quantum many-body problem to a few-body problem in terms of dressed charge and flux excitations. In this work we focus on the one-particle subspace, which contains the decoupled one-charge and one-flux excitation energies.

Often the most efficient way to extract high-order series of one-particle excitation energies is to exploit the linked-cluster theorem and set up a full graph decomposition [51–53]. Here we use an expansion in terms of hypergraphs [54], which has been recently extended to perturbed topological phases allowing to incorporate the non-local mutual anyonic statistics of charges and fluxes into a graph decomposition [47]. We applied the same approach as described in

Ref. [47] to calculate the one-quasi-particle charge and flux excitation energies as well as the ground-state energy per site up to order 10 perturbation theory in the parameters $h_x$ and $h_z$.

The effective Hamiltonian in the single charge and the single flux sector is given by a one-particle hopping problem. As the effective one-charge (one-flux) Hamiltonian is a lattice hopping problem on a honeycomb (triangular) lattice it can be diagonalized by a Fourier transformation.

Let us first consider the one-charge state $|\vec{r}, l; \text{⬥}\rangle$ with $l \in \{1, 2\}$ denoting the two sites of the unit cell located at $\vec{r}$ on the honeycomb lattice. The real space one-particle hopping amplitudes $a^{\text{⬥}}_{\vec{\delta}, lm}$ with $l, m \in \{1, 2\}$ as a high-order series are given by

$$a^{\text{⬥}}_{\vec{\delta}, lm} = \langle \vec{r} + \vec{\delta}, l; \text{⬥}| H_{\text{eff}} - E_0 |\vec{r}, m; \text{⬥}\rangle . \tag{17}$$

The single-charge problem on the honeycomb lattice can then again be simplified in Fourier space by introducing momentum states $|\vec{k}, l; \text{⬥}\rangle \equiv \sqrt{\frac{2}{N_s}} \sum_{\vec{r}} \exp(i\vec{k}\vec{r}) |\vec{r}, l; \text{⬥}\rangle$. This yields a $2 \times 2$ matrix per momentum $\vec{k}$

$$\omega^{\text{⬥}}_{lm}(\vec{k}) \equiv \langle \vec{k}, l; \text{⬥}| H_{\text{eff}} - E_0 |\vec{k}, m; \text{⬥}\rangle , \tag{18}$$

which can be easily diagonalized. One therefore obtains two single-flux bands. The overall minimum of the two bands is called the one-charge gap $\Delta^{\text{⬥}}$. It is located at $\vec{k} = (0, 0)$ for $h_z > 0$ and at $\vec{k} = (\pi, \pi)$ for $h_z < 0$. However, due to the bipartite structure of the honeycomb lattice, one finds that the gap series is identical for negative and positive $h_z$. The explicit series of the ground-state energy per site as well as the charge and flux gap are listed in App. A.

Next we consider the one-flux state $|\vec{r}; \text{⊙}\rangle$. We have calculated the real space one-particle hopping amplitudes $a^{\text{⊙}}_{\vec{\delta}}$ as a high-order series given by

$$a^{\text{⊙}}_{\vec{\delta}} = \langle \vec{r} + \vec{\delta}; \text{⊙}| H_{\text{eff}} - E_0 |\vec{r}; \text{⊙}\rangle . \tag{19}$$

The single-flux problem on the triangular lattice can then be diagonalized in Fourier space by introducing momentum states $|\vec{k}; \text{⊙}\rangle \equiv \sqrt{\frac{1}{N_p}} \sum_{\vec{r}} \exp(i\vec{k}\vec{r}) |\vec{r}; \text{⊙}\rangle$ yielding the one-flux dispersion

$$\omega^{\text{⊙}}(\vec{k}) \equiv \langle \vec{k}; \text{⊙}| H_{\text{eff}} - E_0 |\vec{k}; \text{⊙}\rangle . \tag{20}$$

The minimum of $\omega^{\text{⊙}}(\vec{k})$ is called the one-flux gap $\Delta^{\text{⊙}}$. It is located at $\vec{k} = (0, 0)$ for $h_x > 0$ and at $\vec{k} = \pm(\frac{2\pi}{3}, -\frac{2\pi}{3})$ for $h_x < 0$.

## 3.2 Extrapolation methods

The investigated high-order series in two perturbation parameters $\lambda_1$ and $\lambda_2$ are of the form

$$f_{o_{\max}}(\lambda_1, \lambda_2) = \sum_{j=0}^{o_{\max}} \prod_{k=0}^{j} a_{j,k} \lambda_1^k \lambda_2^{j-k}, \tag{21}$$

where $a_{j,k}$ are real coefficients and $o_{\max}$ is the achieved highest order. In order to obtain a series in a single variable, one can always express $(\lambda_1, \lambda_2)$ as $\lambda(\cos(\phi), \sin(\phi))$, such that Eq. (21) can be rewritten as

$$f_{o_{\max}}(\lambda, \phi) = \sum_{j=0}^{o_{\max}} \lambda^j \prod_{k=0}^{j} a_{j,k} \cos(\phi)^k \sin(\phi)^{j-k}. \tag{22}$$

This function can then be analysed for fixed $\phi$ and different orders $j \in [1, 2, \ldots, o_{\max}]$ such that we are left with only the absolute value $\lambda$ as free variable.

To enhance the convergence radius of the bare series, we use Padé and DLogPadé extrapolations. A detailed introduction of these techniques can be found in [55]. The Padé extrapolation of a function $f_{o,\phi}$ is given by

$$P[L,M]_{f_{o,\phi}}(\lambda) = \frac{P_L(\lambda)}{Q_M(\lambda)} = \frac{p_0 + p_1\lambda^1 + \ldots + p_L\lambda^L}{q_0 + q_1\lambda^1 + \ldots + q_M\lambda^M}. \tag{23}$$

Here $P_L$ and $Q_M$ are polynomials of order $L$ and $M$ with $L + M \leq o$. The coefficients $p_i$ and $q_i$ can be determined uniquely using the condition that for a given order $o$ and at given fixed angle $\phi$, the Taylor expansion of the Padé extrapolant at $\lambda = 0$ must be equivalent to $f_{o,\phi}(\lambda)$.

To describe continuous quantum phase transition displaying algebraic behaviour close to a quantum critical point $\lambda_c$, DLogPadé extrapolation is necessary. Specifically, the gap closing at $\lambda_c$ behaves as

$$\Delta \propto (\lambda - \lambda_c)^{z\nu} \tag{24}$$

close to the critical point, where $z$ is the dynamical exponent and $\nu$ the correlation length exponent [56]. Considering the logarithmic derivative, one obtains

$$\frac{d}{d\lambda}\log[\Delta(\lambda)] = \frac{\Delta'(\lambda)}{\Delta(\lambda)} \propto \frac{z\nu}{(\lambda - \lambda_c)}. \tag{25}$$

Applying a Padé extrapolation to Eq. (25) with $L + M \leq o - 1$ then allows to extract the critical point as the pole and the critical exponent $z\nu$ as the residue. This technique is called DLogPadé extrapolation.

A DLogPadé approximant $[L, M]$ can display unphysical poles. To identify the physically relevant ones, prior knowledge of the system is required, since sometimes unphysical poles appear before the physical ones spoiling the extrapolation. Further, the location of a physical pole and its associated critical exponent can be distorted by other, non-physical poles. To ensure that we only take into account the relevant poles that are not deformed by unphysical ones, we disregard DLogPadé approximants with other poles within a radius $|\lambda| = 0.02$ in the complex plane.

In this work, we calculate the low-field series expansion of the charge and flux gap up to order 10 in the two perturbation parameters $h_x$ and $h_z$. We use DLogPadé approximants $[4,5]$, $[5,4]$, $[3,6]$, $[6,3]$, and $[4,4]$ as best, most diagonal approximants. In the following, we will always determine the average of these approximants with a confidence band that depicts the sample standard deviation to estimate the uncertainty of the extrapolation. We further select the physical poles from all possible poles by using their proximity to already known poles. This is done by extrapolating $f_{10}(h, \phi)$ for $\phi = \{0, \frac{\pi}{2}, \pi\}$ and adding/subtracting $\Delta\phi = 0.005$. For every increment, the pole closest to the pole selected before is chosen as the physical pole. However, in certain field directions, it can happen for specific DLogPadé approximants that poles deviate from the other DLogPadé approximants, which originates from other non-physical poles in the complex plane.

## 3.3 Stochastic series expansion quantum Monte Carlo

As we will demonstrate in Sec. 4.2, it is possible to map the charge-free sector of the toric code on the honeycomb lattice in a negative magnetic field on the frustrated antiferromagnetic transverse-field Ising model (TFIM) on the triangular lattice. Therefore, we are interested in the ground-state energy of this model to infer the lowest energy of the charge-free sector. To achieve this objective, we apply the stochastic series expansion (SSE) quantum Monte Carlo technique (QMC) technique taylored for this specific model by S. Biswas et al. [57, 58].

The Hamiltonian of the antiferromagnetic TFIM is given by

$$H = J \sum_{\langle i,j \rangle} \sigma_i^z \sigma_j^z - h \sum_i \sigma_i^x \qquad (26)$$

with Pauli matrices $\sigma_i^{x/z}$ describing spin $1/2$ degrees of freedom at position $\vec{r}_i$, the strength of the Ising interaction $J > 0$, the transverse-field strength $h$, and the sum over nearest-neighboring sites $\langle i,j \rangle$ [59–61]. Regarding the spectrum of the antiferromagnetic Ising model in the absence of a transverse field, one observes an extensive ground-state degeneracy due to geometric frustration [62,63]. This originates from the fact that it is not possible to align the spins on a triangle in a way to optimize all antiferromagnetic couplings. A single decoupled triangle with antiferromagnetic Ising interactions has six degenerate ground states (three configurations with two spins up one down and three with two down one up) and two degenerate excited states (all spins equally oriented).

The competing nature of the antiferromagnetic interaction in the regarded model renders the well esablished SSE QMC sampling of general TFIMs introduced by A. Sandvik [64] impractical due to large autocorrelation times between configurations in the Monte Carlo time [57, 58].

The SSE approach is based on a high-temperature expansion of the partition function

$$Z = \text{Tr}\{e^{-\beta H}\} = \sum_{n=0}^{\infty} \sum_{\{|\alpha\rangle\}} \frac{(-\beta)^n}{n!} \langle \alpha | H^n | \alpha \rangle \ . \qquad (27)$$

The key idea is to extend the configuration space in imaginary time by using an adequate decomposition $H = -\sum_i H_i$ to rewrite

$$H^n = (-1)^n \sum_{S_n} \prod_{p=1}^n H_i \qquad (28)$$

as a sum of sequences $S_n$ of the operators $H_i$ [64–67]. The SSE method approximates (27) by neglecting operator sequences $S_n$ longer than some appropriately chosen $\mathcal{L}$ [64, 65]. Sequences with less than $\mathcal{L}$ operator-index pairs are padded to length $\mathcal{L}$ by randomly inserting identity operators [64–67]. This leads to an efficient sampling scheme with an exponentially small error [64]. The configuration space to be sampled by a Markov chain Monte Carlo (MCMC) is $\mathcal{C} = \{S_{\mathcal{L}}\} \times \{|\alpha\rangle\}$. In the general approach to TFIMs the Hamiltonian is decomposed into Ising bond operators $|J| - J\sigma_i^z \sigma_j^z$, transverse-field operators $h\sigma_i^x$ and further physically not relevant operators required for algorithmic reasons [64]. The MCMC sampling is then performed by repeating diagonal update steps followed by off-diagonal quantum cluster updates [64]. In the diagonal update, step operators $H_i$, which are diagonal in the computational basis, are exchanged in the sequence with identity operators and vice versa following a Metropolis-Hastings scheme [64]. In the offdiagonal update, quantum clusters in space and imaginary time are constructed which can be flipped without changing the weight of the current configuration [64]. The major development by S. Biswas et al. [57, 58] is to decompose the Ising interaction not into bonds, but into operators associated with the triangles of the triangular lattice

$$J \sum_{\langle i,j \rangle} \sigma_i^z \sigma_j^z = \frac{J}{2} \sum_{\kappa \in \{\Delta, \nabla\}} \left( \sigma_{\kappa,1}^z \sigma_{\kappa,2}^z + \sigma_{\kappa,2}^z \sigma_{\kappa,3}^z + \sigma_{\kappa,3}^z \sigma_{\kappa,1}^z \right) , \qquad (29)$$

where $\kappa \in \{\Delta, \nabla\}$ runs over all triangles of the lattice and $\{1, 2, 3\}$ denotes the spins within the triangle. This decomposition allows for the formulation of an off-diagonal quantum cluster

update which copes with the frustration induced state structure on the triangles in a natural way [57, 58]. With this taylored quantum cluster update the correlation between configurations in Monte Carlo time is reduced substantially and an efficient Monte Carlo sampling is enabled [57, 58]. In this work, we will not provide any further description of the algorithm, since we follow precisely the scheme described in Refs. [57, 58]. For details on the algorithmic design we recommend the Refs. [57, 58, 64, 67].

The SSE method is a finite-temperature finite-system QMC technique. To obtain a ground-state sampling, the temperature needs to be sufficiently low such that the contribution of excited states to the averaged observables is negligible. We follow the systematic approach described in Ref. [68] to ensure the convergence in temperature within the statistical Monte Carlo error. Therefore, the measured observables are at effectively zero temperature.

Using the SSE QMC sampling we measure the mean energy which corresponds to the ground-state energy, if measured at zero temperature [67]. We perform the SSE QMC simulations on $L \times L$ patches of the triangular lattice with periodic boundary conditions for $L \in \{12, 18, 24, 30\}$ at temperatures of $\beta/J = 256$. Further, we extrapolate the ground-state energies to the thermodynamic limit (see Fig. 4). With this procedure we are able to determine estimates for the ground-state energies of the antiferromagnetic TFIM on the triangular lattice in the thermodynamic limit.

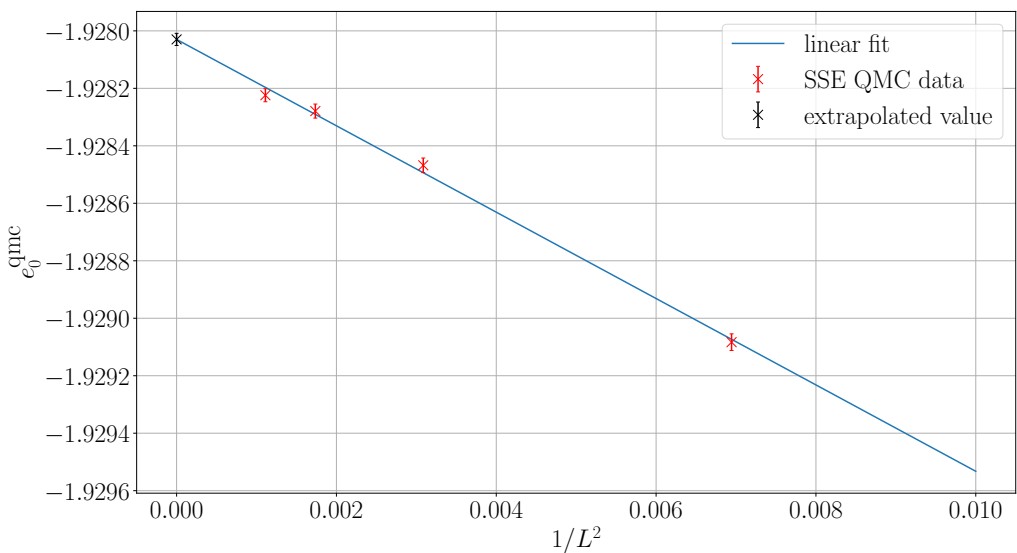

Figure 4: Extrapolation of the ground-state energy per site $e_0^{\mathrm{qmc}}$ for the antiferromagnetic TFIM on the triangular lattice at a transverse-field $h = 1.6$ and Ising coupling $J = 1$.

## 4 Phase transitions for single parallel fields

In this section we focus on the two single-field cases $\vec{h} = (0, 0, h_z)$ and $\vec{h} = (h_x, 0, 0)$ for the toric code on the honeycomb lattice. These field directions are special because in each case (i) one of the two types of stabilizer operators still commutes with the Hamiltonian and (ii) exact duality mappings to TFIMs exist for specific subspaces. Together with our results from series expansion and quantum Monte Carlo simulations, this allows the quantitative calculation of

the quantum phase diagram in these field directions.

## 4.1 Magnetic field in $z$-direction

Setting $\vec{h} = (0, 0, h_z)$, the Hamiltonian (1) reads

$$H_z^{\text{tc}} = -\frac{1}{2}\sum_{\star} X_{\star} - \frac{1}{2}\sum_{\circ} Z_{\circ} - h_z \sum_i \sigma_i^z. \tag{30}$$

In this case, the plaquette operators still commute with the Hamiltonian and their eigenvalues remain conserved quantities. The low-energy physics of both the low-field topological phase of the toric code as well as of the high-field polarized phase is contained in the flux-free sector with $z_{\circ} = +1$ for all $\circ$. This can be rigorously shown in the limit that fluxes cost infinite energy. In this subspace, $H_z^{\text{tc}}$ reduces to

$$H_z^{\text{tc,fluxfree}} = -\frac{N_p}{2} - \frac{1}{2}\sum_{\star} X_{\star} - h_z \sum_i \sigma_i^z. \tag{31}$$

By considering $x_{\star} = \pm 1$ as eigenvalue of a pseudo-spin 1/2 Pauli matrix $\mu_{\star}^z$ located on the vertices of the honeycomb lattice, one finds the non-local duality mapping of $H_z^{\text{tc,fluxfree}}$ to the TFIM on the honeycomb lattice

$$H_{\text{honeycomb}}^{\text{tfim}} = -\frac{1}{2}\sum_{\star}\mu_{\star}^z - h_z \sum_{\langle \star, \star' \rangle} \mu_{\star}^x \mu_{\star'}^x \tag{32}$$

as illustrated on the left side of Fig. 5. The second sum is over nearest-neighbor stars. We stress that the mapping does not hold for degeneracies. Since the honeycomb lattice is bipartite, the TFIM is symmetric under sign change of $h_z$ by an appropriate sublattice rotation. As a consequence, the toric code in a $z$-field realizes a second-order quantum phase transition in the 3D Ising* universality class [69,70] between the topologically ordered and the polarized phase. The quantum critical point is located at $h_{z,c} = 0.234467(5)$ which was determined by cluster Monte Carlo simulations in Ref. [71].

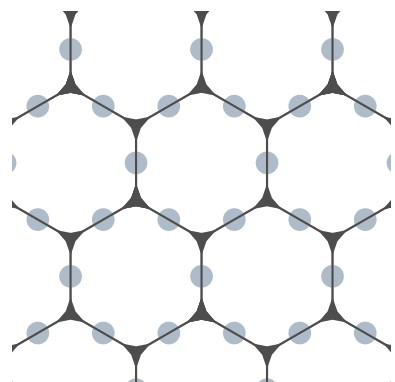 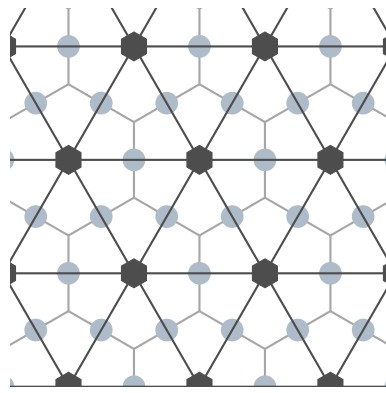

Figure 5: Left: Dual lattice of the stars, which are located on the vertices of the honeycomb lattices and form an honeycomb lattice themselves. Right: Dual lattice of the plaquettes, which are located on the faces of the honeycomb lattice and form a triangular lattice.

In order to gauge the quality of the low-field expansion, we performed DLogPadé extrapolation of the charge gap (see Subsec. 3.2 and App. A for the series). Note that to the best of our

knowledge, for the TFIM on the honeycomb lattice the series of the high-field gap has not yet been calculated. The critical field strength $h_{z,c}$ and the critical exponent $z\nu$ extracted from various DLogPadé extrapolants are listed in Tab. 1 in App. B. Averaging over highest-order extrapolants, yields $h_{z,c} = 0.2352(9)$ and $z\nu = 0.653(10)$ fully consistent with $h_{z,c} = 0.234467(5)$ from quantum Monte Carlo Simulations of the TFIM on the honeycomb lattice and the expected phase transition in the the 3D Ising$^\star$ universality class $z\nu = 0.629971(4)$ [72]. Let us note that a slight overestimation of the critical exponent is well known from extrapolations of high-order series. The quantum phase diagram of the toric code on the honeycomb lattice in a $z$-field therefore consists of the topologically ordered phase for $|h_z| < h_{z,c}$ and a high-field polarized phase separated by a second-order quantum phase transition in the 3D Ising$^\star$ universality class.

## 4.2 Magnetic field in $x$-direction

Setting $\vec{h} = (0, 0, h_x)$, the Hamiltonian (1) can be written as

$$H_x^{\text{tc}} = -\frac{1}{2}\sum_{\lambda} X_{\lambda} - \frac{1}{2}\sum_{\circ} Z_{\circ} - h_x \sum_i \sigma_i^x . \tag{33}$$

In this case the eigenvalues of star operators remain conserved quantities. However, the physics depends strongly on the sign of $h_x$. In the following we therefore discuss both cases separately.

For positive $h_x > 0$, a similar line of arguments can be made as for the $z$-field. The low-energy physics is contained in the charge-free sector with $x_{\lambda} = +1$ for all $\lambda$ so that $H_x^{\text{tc}}$ reduces to

$$H_x^{\text{tc,chargefree}} = -\frac{N_s}{2} - \frac{1}{2}\sum_{\circ} Z_{\circ} - h_x \sum_i \sigma_i^x \tag{34}$$

in this subspace. Again considering $z_{\circ} = \pm 1$ as eigenvalue of a pseudo-spin 1/2 Pauli matrix $\mu_{\circ}^z$ located on the plaquette centers of the honeycomb lattice, one finds the non-local duality mapping of $H_x^{\text{tc,chargefree}}$ to the ferromagnetic TFIM

$$H_{\text{triangular}}^{\text{tfim}} = -\frac{1}{2}\sum_{\circ} \mu_{\circ}^z - h_x \sum_{\langle \circ, \circ' \rangle} \mu_{\circ}^x \mu_{\lambda'}^x . \tag{35}$$

on the triangular lattice as illustrated on the right side of Fig. 5. For positive $h_x$, the toric code in an $x$-field is therefore isospectral to the ferromagnetic TFIM on a triangular lattice, which again realizes a quantum phase transition in the 3D Ising$^*$ universality class. The quantum critical point is located at $h_{x,c} = 0.104863(2)$ obtained from quantum Monte Carlo simulations [71]. Our results are consistent with the high-field expansion of the ferromagnetic TFIM on the triangular lattice by He et al. [73].

As for the $z$-field case, we gauge the quality of the low-field expansion by performing DLogPadé extrapolation of the flux gap (see App. A for the series). The critical field strength $h_{x,c}$ and the critical exponent $z\nu$ extracted from various DLogPadé extrapolants are listed in Tab. 2 in App. B. Averaging over highest-order extrapolants yields $h_{x,c}^+ = 0.10491(13)$ and $z\nu = 0.6404(11)$ fully consistent with $h_{x,c}^+ = 0.104863(2)$ from quantum Monte Carlo simulations of the ferromagnetic TFIM on the triangular lattice and the expected phase transition belonging to the 3D Ising universality class $z\nu = 0.629971(4)$ [72]. Let us note again that a slight overestimation of the critical exponent is well known from extrapolations of high-order series.

For $h_x < 0$, the situation is completely different. First, the high-field polarized phase is not in the charge-free sector, but in the charge-full sector with $x_{\lambda} = -1$ for all $\lambda$. Second, the

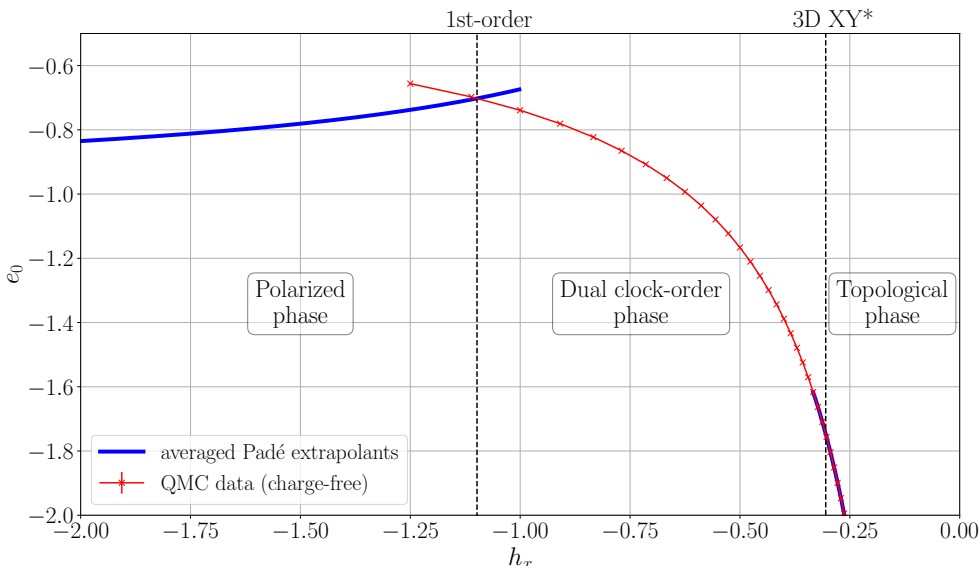

Figure 6: Comparison of the ground-state energy for different phases and charge-sectors. Blue lines show the average of Padé extrapolants of order 10 for the topological phase from the low-field expansion and the fully polarized phase from the high-field expansion. The ground-state energy of the charge-free and polarized (charge-full) sector intersect at $h_{x,1st}^{-} = -1.0982(1)$.

analogue duality mapping in the charge-free sector yields the antiferromagnetic TFIM on the triangular lattice (Hamiltonian (35) with $h_x < 0$). This highly frustrated model is known to realize a quantum phase transition in the 3D XY universality class separating the polarized phase from the three-sublattice ordered phase resulting from an order by disorder scenario. Consequently, we coin the associated phase in the original formulation of the toric code in this field direction *dual clock-order phase*. The quantum critical point is located at $h_c^- = -0.303(9)$ obtained from quantum Monte Carlo simulations [61] and high-order series expansion [74].

Next we investigate whether the 3D XY$^\star$ quantum phase transition in the flux-free sector is realized in the phase diagram of the toric code in the $x$-field or whether the first-order phase transition to the charge-full polarized phase takes place first when decreasing $h_x$ from zero to $-\infty$. To this end we compare the ground-state energy per site $e_0^{qmc}$ of the transverse-field Ising model on the full parameter axis obtained by quantum Monte Carlo simulations (see Subsec. 3.3) to the ground-state energy per site $e_0^{hf}$ from high-order series expansion about the high-field polarized limit. The crossing point of both energies corresponds to the first-order phase transition between the charge-free and charge-full sector. It can then be compared to the quantum critical point $h_{x,c}^-$ of the 3D XY$^\star$ transition in the charge-free sector.

Our findings are shown in Fig. 6. This clearly indicates that the first-order phase transition occurs for $h_{x,1st}^- = -1.0982(1) < h_{x,c}^-$ so that the 3D XY$^\star$ quantum phase transition is part of the quantum phase diagram. Let us note that one can tune the first-order phase transition between the charge-free and charge-full sector by introducing anisotropic couplings in front of star and plaquette operators. Concretely, if one decreases the coupling of star operators from $-1/2$ to large negative values, then the energetic costs of charges become also large and therefore the first-order phase transition shifts to large negative $h_x$ values. We further stress that this analysis was restricted to the charge-free and the charge-full sector. It would certainly

be interesting to also investigate the other sectors.

The 3D XY quantum phase transition of the antiferromagnetic TFIM is expected for $h^-_{x,c} = 0.303(9)$ with $z\nu = 0.67169(7)$ [75,76]. The critical field strength $h^-_{x,c}$ and the critical exponent $z\nu$ extracted from various DLogPadé extrapolants are listed in Tab. 3 in App. B. Averaging over highest-order extrapolants gives $h^-_{z,c} = 0.3059(4)$ and $z\nu = 0.720(5)$. One notices an enhanced uncertainty of the extrapolation around the expected values. This can be traced back to the slower convergence of the DLogPadé extrapolants due to the alternating sign structure of the series of the flux gap (see App. A).

The quantum phase diagram of the toric code on the honeycomb lattice in an $x$-field is therefore richer than the one in the $z$-field as well as the one of the toric code in a single parallel field on the square lattice. For $h_x > 0$, one has a single second-order phase transition in the 3D Ising$^\star$ universality class at $h^+_{x,c}$ separating the low-field topological phase from the high-field polarized phase. For $h_x < 0$, there is a second-order phase transition in the 3D XY$^\star$ universality class at $h^-_{x,c}$ between the topological phase and a phase with three-sublattice order and a first-order phase transition at $h^-_{x,1st} < h^-_{x,c}$ to the high-field polarized phase.

## 5   Phase transitions out of the topological phase in the $xz$-plane

In this part we investigate the quantum robustness of the topological phase in the full parallel field plane $h = (h_x, 0, h_z)$. Here, no exact mappings can be exploited. We therefore focus on the low-field expansion of the charge and flux gap and study the gap-closing transitions by DLogPadé extrapolation. As shown in the last section, this yields satisfactory results in all single-field cases. We therefore extrapolate the charge and the flux gap for all field directions and analyse which gap for which momentum closes first, which locates the breakdown of the topological phase. Our results for the extension of the topological phase in the $xz$-plane are shown in Fig. 7 and the corresponding results for the critical exponent $z\nu$ are plotted in Fig. 8.

The extension of the topological phase is symmetric with respect to $h_z \leftrightarrow -h_z$ while it is asymmetric for $h_x \leftrightarrow -h_x$. The upper and lower critical line include the single-field cases for $h_x = 0$ where a closing of the charge gap and a second-order 3D Ising$^\star$ phase transition is known. Because the critical exponent $z\nu \approx 0.64$ is essentially constant on this critical line (see domain (I) in Fig. 8), we associate a 3D Ising$^\star$ universality class on the whole line in full analogy to the well studied toric code in a parallel field on the square lattice [25, 30]. The same universality class is present on the right critical line for $h_x > 0$ (see domain (II) in Fig. 8). On this line the flux gap with momentum $\vec{k} = (0, 0)$ is closing, which includes also the single-field case $h_z = 0$ where the low-energy physics is described by the ferromagnetic TFIM on the triangular lattice. In contrast, on the left critical line with $h_x < 0$, the flux gap closes at momenta $\vec{k} = \pm(\frac{2\pi}{3}, -\frac{2\pi}{3})$. This line includes the case $h_z = 0$, where the low-energy physics corresponds to the frustrated antiferromagnetic TFIM on the triangular lattice and where the quantum phase transition is in the 3D XY$^\star$ universality class. Let us note that the location of the first-order phase transition to the charge-full sector takes place at smaller values of $h_x$. Again, the critical exponent $z\nu$ obtained from various DLogPadé extrapolants is essentially constant along this line so that the quantum phase transition is expected to be 3D XY$^\star$ on this parameter line.

Notably, the values of the critical exponent $z\nu$ deviate from the expected values for 3D Ising$^\star$ and 3D XY$^\star$ at all crossing points, where the charge and the flux gap close simultaneously. One has to distinguish two situations: First, there are crossing points where two critical lines belonging to the (same) 3D Ising$^\star$ universality class come together. Here the flux and the charge gap close simultaneously for the momenta $\vec{k} = (0, 0)$ or $\vec{k} = (\pi, \pi)$. This situation is

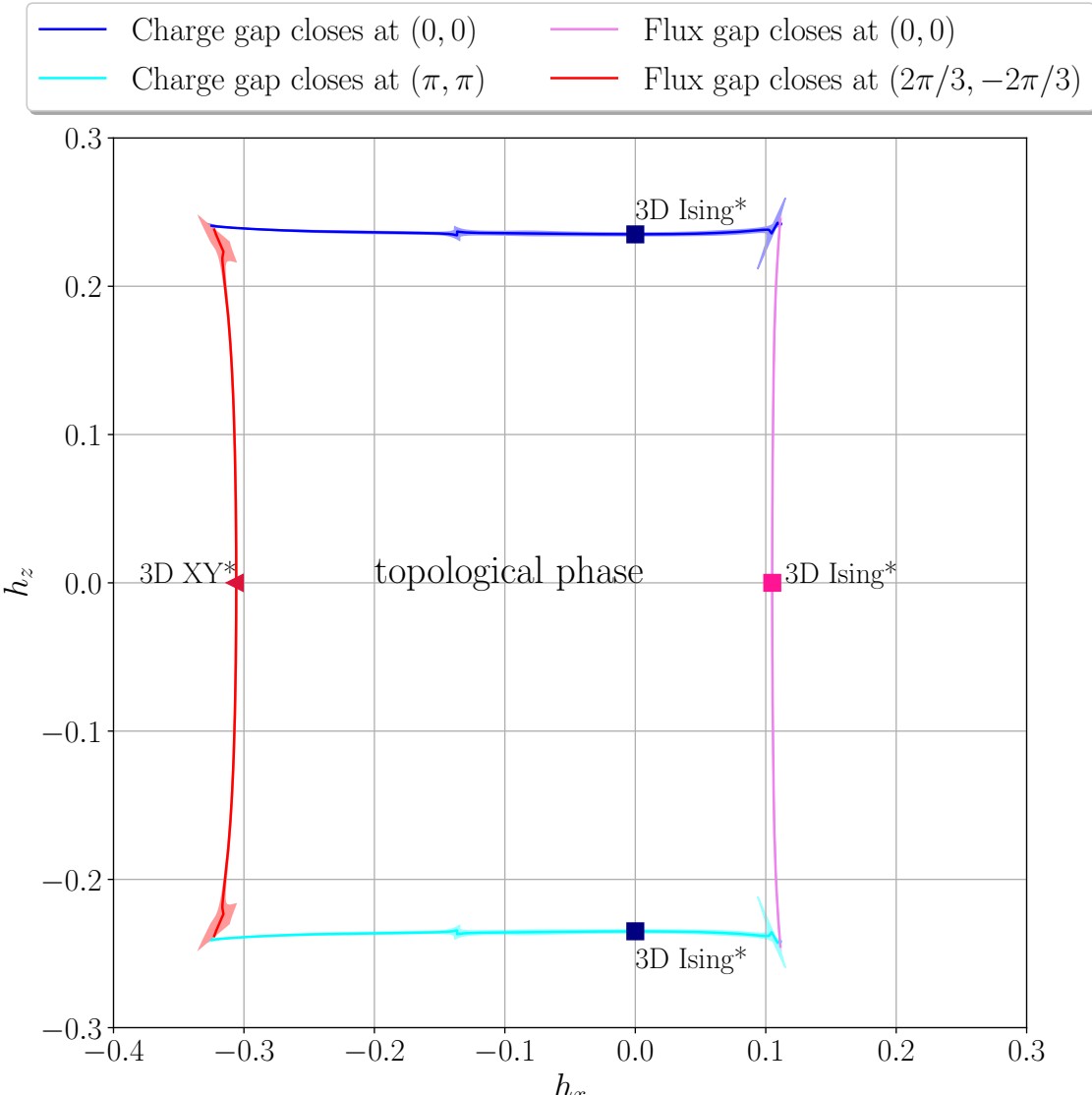

Figure 7: Extension of the topological phase as a function of $h_x$ and $h_z$. Solid lines correspond to the gap closing obtained from DLogPadé extrapolation of the flux and charge gap including standard deviation as a shaded confidence band. Quantum phase transitions at $h_x = 0$ are 3D Ising* transitions known from the duality mapping to the TFIM on the unfrustrated honeycomb lattce. Quantum phase transitions at $h_z = 0$ are 3D Ising* (3D XY*) transitions known from the duality mapping to the ferromagnetic (antiferromagnetic) TFIM on the triangular lattice for $h_x > 0$ ($h_x < 0$).

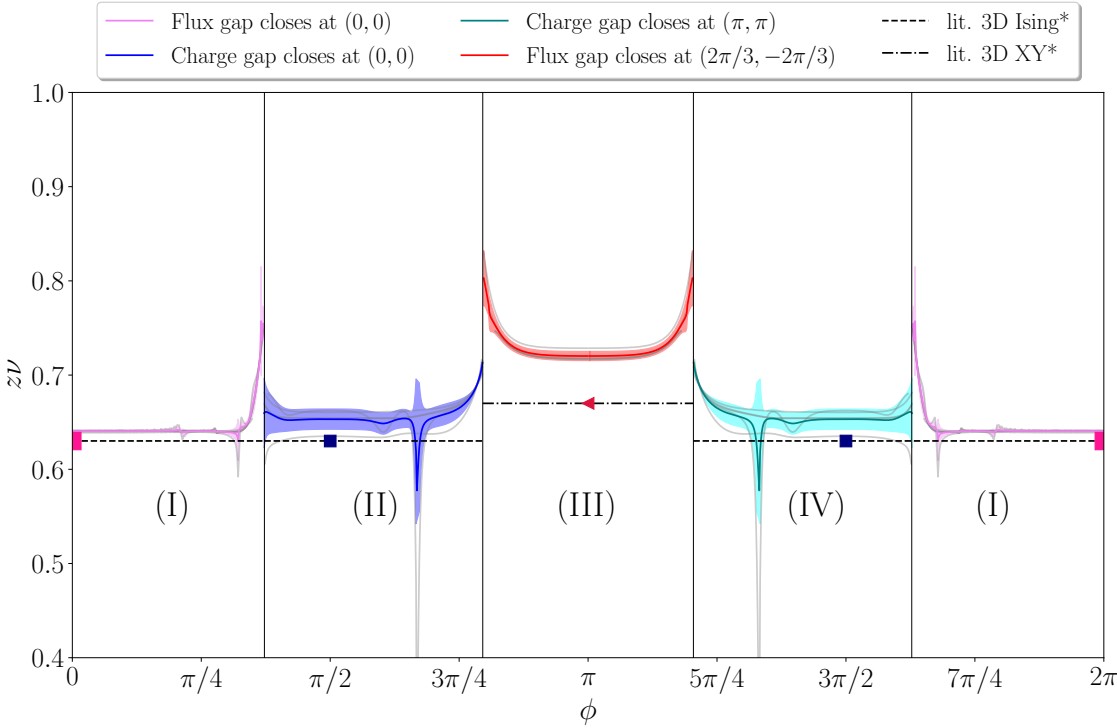

Figure 8: Averaged critical exponent $z\nu$ plotted in color against the angle $\phi$ parametrizing the magnetic field as $\vec{h} = |h|(\cos(\phi), 0, \sin(\phi))$. The horizontal dashed lines indicate the literary values for the critical exponent $z\nu$ of the 3D Ising$^\star$ and 3D XY$^\star$ universality class, taken from [72] and [75,76], respectively. The light grey lines depict the individual DLogPadé approximants used in the averaging. Vertical lines separate the domains (I)-(IV). Domain (I) is associated with the closing of the flux gap at $\vec{k} = (0,0)$ corresponding to a 3D Ising$^\star$ quantum phase transition. Domain (II) is associated with the closing of the charge gap at $\vec{k} = (0,0)$ corresponding to a 3D Ising$^\star$ quantum phase transition. Domain (III) is associated with the closing of the flux gap at $\vec{k} = \pm(\frac{2\pi}{3}, -\frac{2\pi}{3})$ corresponding to a 3D XY$^\star$ transition. Domain (IV) is associated with the closing of the flux gap at $\vec{k} = (\pi, \pi)$ corresponding to a 3D Ising$^\star$ quantum phase transition. The boundaries between different domains are associated with potential multicritical points.

similar to the one of the toric code in a parallel field on the square lattice [25,30]. However, in the latter case the model possesses a supersymmetry so that the crossing point is exactly located on the parameter lines $h_x = \pm h_z$. On the honeycomb lattice this supersymmetry is not present. It is therefore remarkable that the critical exponent from series expansion is $z\nu \approx 0.75$ and therefore very similar to the square lattice case. We therefore conjecture that these crossing points are multi-critical points belonging to the same universality class as the ones on the square lattice for $h_x = \pm h_z$. Second, there are crossing points where a 3D Ising$^\star$ and a 3D XY$^\star$ critical line merge corresponding to a simultaneously closing of the charge and flux gap at different momenta. We note that the critical exponents from the extrapolation of the two series do not agree ($z\nu \approx 0.7$ for the flux gap and $z\nu \approx 0.8$ for the charge gap). This behavior of the extrapolation is very interesting, but certainly deserves a deeper understanding which is beyond the capabilities of the series expansion.

# 6 Conclusions

In this work we have calculated the quantum robustness of the topological phase for the toric code on the honeycomb lattice in the presence of a uniform parallel field. For the single-parallel field cases in $x$- and $z$-direction duality transformations allow quantitative insights of the full quantum phase diagram. One finds a second-order quantum phase transition in the 3D Ising$^\star$ universality class for a $z$-field independent of the sign of the field separating the topological phase from the high-field polarized phase. The same is true for positive fields in $x$-direction where an analogue mapping in the charge-free sector yields a ferromagnetic TFIM on the triangular lattice. In contrast, for negative $x$-fields the low-energy physics of the charge-free sector is given by the highly frustrated antiferromagnetic TFIM on the triangular lattice displaying a 3D XY$^\star$ phase transition. However, the charge-free sector does not contain the high-field polarized phase for negative $x$-fields and a first-order phase transition to the polarized phase in the charge-full sector takes place at larger negative field values, which can be quantitatively located by comparing quantum Monte Carlo simulations and high-field series expansions. At this point it should be stressed, that we only considered the charge-free and charge-full sectors, while all other sectors of fractional filling were neglected. A more thorough analysis is needed, since the tools presented in this work do not allow for an examination of those sectors. The full extension of the topological phase in the $xz$-field plane can then be determined by low-field series expansions analysing the closing of the charge and/or flux gap. One obtains critical lines of 3D Ising$^\star$ and 3D XY$^\star$ type. The crossing points between the two universality classes correspond to the simultaneous closing of both gaps. Here the extrapolation of the gap series reveals enhanced values for the critical exponents $z\nu$ indicating different quantum-critical properties. This is similar to the same findings for the toric code in a parallel field on the square lattice [25, 30, 38].

In an next step it would be interesting to explore the full quantum phase diagram also outside the topological phase. One interesting question is the evolution of the first-order point on the $h_z = 0$ axis and $h_x < 0$ in the $xz$-field plane. Another relevant aspect is the inclusion of a transverse field $h_y$ and its influence of the topological phase as well as the associated phase transitions. Overall, as for the toric code on the dice lattice [31], this shows that geometric frustration can induce interesting novel properties in topological phases. It would be interesting to apply quantum Monte Carlo simulations or tensor network calculations to get a deeper understanding of these quantum-critical properties. This would also allow to investigate whether other sectors different to the charge-free and charge-full sector are relevant for the quantum phase diagram at negative $x$-fields.

## Acknowledgements

We acknowledge the valuable input and fruitful discussions with Julien Vidal. The authors thankfully acknowledge the scientific support and HPC resources provided by the Erlangen National High Performance Computing Center (NHR@FAU) of the Friedrich-Alexander-Universität Erlangen-Nürnberg (FAU).

**Funding information**    KPS acknowledges financial support by the German Science Foundation (DFG) through the grant SCHM 2511/11-1 and the Munich Quantum Valley, which is supported by the Bavarian state government with funds from the Hightech Agenda Bayern Plus. JAK and KPS further acknowledge support through the TRR 306 QuCoLiMa ("Quantum Cooperativity of Light and Matter") - Project-ID 429529648. The hardware of NHR@FAU is funded by the German Research Foundation DFG.

# A    Series from the low-field expansion

This appendix contains the bare series of the ground-state energy per site as well as the charge and flux gap obtained from low-field series expansions. The ground-state energy per site $e_0^{\text{lf}} = E_0/N$ reads

$$
\begin{aligned}
e_0^{\text{lf}} = &-\frac{1}{2} - \frac{1}{2}h_z^2 - \frac{3}{8}h_z^4 - \frac{61}{16}h_z^6 - \frac{2401}{128}h_z^8 - \frac{683177}{3072}h_z^{10} - \frac{1}{2}h_x^2 + \frac{1}{4}h_x^2h_z^2 + \frac{5}{8}h_x^2h_z^4 \\
&+ \frac{8489}{768}h_x^2h_z^6 + \frac{211421}{3072}h_x^2h_z^8 + h_x^3 - \frac{3}{2}h_x^3h_z^2 - \frac{257}{64}h_x^3h_z^4 - \frac{341117}{4608}h_x^3h_z^6 - \frac{29}{8}h_x^4 \\
&+ \frac{219}{32}h_x^4h_z^2 + \frac{2467}{128}h_x^4h_z^4 + \frac{13528841}{36864}h_x^4h_z^6 + \frac{33}{2}h_x^5 - \frac{2533}{64}h_x^5h_z^2 - \frac{51453}{512}h_x^5h_z^4 \\
&- \frac{713}{8}h_x^6 + \frac{197059}{768}h_x^6h_z^2 + \frac{2071729}{3456}h_x^6h_z^4 + \frac{2105}{4}h_x^7 - \frac{510785}{288}h_x^7h_z^2 - \frac{426679}{128}h_x^8 \\
&+ \frac{1422176993}{110592}h_x^8h_z^2 + \frac{1421167}{64}h_x^9 - \frac{118233091}{768}h_x^{10}.
\end{aligned}
\tag{36}
$$

The resulting charge gap $\Delta^\star$ at momentum $\vec{k} = (0,0)$ of the lowest band reads

$$
\begin{aligned}
\Delta^\star = &\; 1 + 3h_x^3 + 15h_x^4 + 102h_x^5 + \frac{2961}{4}h_x^6 + 5547h_x^7 + \frac{1375959}{32}h_x^8 + \frac{21821799}{64}h_x^9 \\
&+ \frac{705563465}{256}h_x^{10} - 3h_z + \frac{3}{2}h_x^2h_z + \frac{141}{8}h_x^4h_z + 72h_x^5h_z + 543h_x^6h_z + \frac{29331}{8}h_x^7h_z \\
&+ \frac{3444063}{128}h_x^8h_z + \frac{51377529}{256}h_x^9h_z - \frac{3}{2}h_z^2 + \frac{3}{4}h_x^2h_z^2 + \frac{123}{8}h_x^3h_z^2 - \frac{69}{32}h_x^4h_z^2 + \frac{12801}{128}h_x^5h_z^2 \\
&- \frac{95097}{256}h_x^6h_z^2 - \frac{8835809}{1536}h_x^7h_z^2 - \frac{1390564685}{18432}h_x^8h_z^2 - 3h_z^3 + \frac{51}{8}h_x^2h_z^3 + \frac{117}{16}h_x^3h_z^3 \\
&+ \frac{15423}{64}h_x^4h_z^3 + \frac{68475}{128}h_x^5h_z^3 + \frac{16592129}{3072}h_x^6h_z^3 + \frac{4149521}{144}h_x^7h_z^3 - \frac{81}{8}h_z^4 + \frac{507}{32}h_x^2h_z^4 \\
&+ \frac{21681}{128}h_x^3h_z^4 + \frac{59511}{256}h_x^4h_z^4 + \frac{5762129}{1536}h_x^5h_z^4 + \frac{35911777}{3072}h_x^6h_z^4 - \frac{57}{2}h_z^5 + \frac{10251}{128}h_x^2h_z^5 \\
&+ \frac{41421}{128}h_x^3h_z^5 + \frac{3158621}{1024}h_x^4h_z^5 + \frac{16365285}{2048}h_x^5h_z^5 - \frac{1011}{16}h_z^6 + \frac{78547}{512}h_x^2h_z^6 \\
&+ \frac{2267399}{1536}h_x^3h_z^6 + \frac{94203515}{36864}h_x^4h_z^6 - \frac{8529}{32}h_z^7 + \frac{984011}{1024}h_x^2h_z^7 + \frac{5386491}{1024}h_x^3h_z^7 \\
&- \frac{57357}{64}h_z^8 + \frac{5072015}{1536}h_x^2h_z^8 - \frac{847545}{256}h_z^9 - \frac{10139367}{1024}h_z^{10}.
\end{aligned}
\tag{37}
$$

For $h_x > 0$, the flux gap $\Delta_{\text{pos}}^\circ \equiv \omega^\circ((0,0))$ reads

$$
\begin{aligned}
\Delta_{\text{pos}}^\circ = &\; 1 - 6h_x - 12h_x^2 - 42h_x^3 - 252h_x^4 - \frac{3153}{2}h_x^5 - \frac{44379}{4}h_x^6 - \frac{2570661}{32}h_x^7 - \frac{9821055}{16}h_x^8 \\
&- \frac{1222762161}{256}h_x^9 - \frac{39126191841}{1024}h_x^{10} + 3h_xh_z^2 + \frac{21}{2}h_x^2h_z^2 + \frac{543}{8}h_x^3h_z^2 + \frac{3795}{8}h_x^4h_z^2 \\
&+ \frac{500301}{128}h_x^5h_z^2 + \frac{8255405}{256}h_x^6h_z^2 + \frac{422330345}{1536}h_x^7h_z^2 + \frac{5509372663}{2304}h_x^8h_z^2 + \frac{51}{4}h_xh_z^4 \\
&+ \frac{315}{16}h_x^2h_z^4 + \frac{18687}{64}h_x^3h_z^4 + \frac{67947}{64}h_x^4h_z^4 + \frac{6069569}{512}h_x^5h_z^4 + \frac{1358138455}{18432}h_x^6h_z^4 + \frac{63}{4}h_z^6 \\
&+ \frac{345}{4}h_xh_z^6 + \frac{192675}{256}h_x^2h_z^6 + \frac{1526337}{512}h_x^3h_z^6 + \frac{997953467}{36864}h_x^4h_z^6 + \frac{153}{2}h_z^8 \\
&+ \frac{64425}{64}h_xh_z^8 + \frac{55903333}{12288}h_x^2h_z^8 + \frac{53511}{32}h_z^{10}.
\end{aligned}
\tag{38}
$$

| L\M | 1 | 2 | 3 | 4 | 5 | 6 | 7 | 8 |
|-----|-----|-----|-----|-----|-----|-----|-----|-----|
| 1 | 0.24242424 | 0.25781969 | 0.23200802 | 0.23080061 | 0.23679399 | 0.23475628 | 0.23440245 | 0.23426491 |
| 2 | 0.22916667 | 0.23148168 | 0.23715 | 0.23498765 | 0.23569037 | 0.23102955 | 0.23425861 | |
| 3 | 0.23197745 | 0.22792844 | 0.23489925 | 0.23545749 | 0.23529536 | 0.23494875 | | |
| 4 | 0.24172508 | 0.23721554 | 0.23573311 | 0.23528599 | 0.23658567 | | | |
| 5 | 0.23346587 | 0.23429127 | 0.23302091 | 0.234935 | | | | |
| 6 | 0.23438328 | 0.23386286 | 0.23408175 | | | | | |
| 7 | 0.23318416 | 0.23410348 | | | | | | |
| 8 | 0.23717572 | | | | | | | |

(a)

| L\M | 1 | 2 | 3 | 4 | 5 | 6 | 7 | 8 |
|-----|-----|-----|-----|-----|-----|-----|-----|-----|
| 1 | 0.70523416 | 0.75134165 | 0.62558181 | 0.614496957 | 0.68549843 | 0.64861508 | 0.641914030 | 0.639186402 |
| 2 | 0.59574382 | 0.61909147 | 0.68243442 | 0.65659783 | 0.66700772 | 0.55522007 | 0.639042419 | |
| 3 | 0.62551367 | 0.587570337 | 0.65534519 | 0.66300541 | 0.660544366 | 0.65435025 | | |
| 4 | 0.76845162 | 0.69232431 | 0.66773007 | 0.660382412 | 0.66232032 | | | |
| 5 | 0.62377308 | 0.639149164 | 0.61067312 | 0.65401315 | | | | |
| 6 | 0.65401315 | 0.630628696 | 0.635124677 | | | | | |
| 7 | 0.61535906 | 0.635619664 | | | | | | |
| 8 | 0.71691764 | | | | | | | |

(b)

Table 1: Results from the DLogPadé extrapolation of the charge gap at $\vec{k} = (0,0)$ for a single parallel field $\vec{h} = (0,0,h_z)$. (a) displays the critical field strength $h_{z,c}$ while (b) shows the corresponding critical exponent.

For $h_x < 0$, the flux gap $\Delta^\circ_{\text{neg}} \equiv \omega^\circ(\pm(\frac{2\pi}{3}, -\frac{2\pi}{3}))$ reads

$$
\begin{aligned}
\Delta^\circ_{\text{neg}} = {} & 1 + 3h_x + \frac{3}{2}h_x^2 + \frac{15}{2}h_x^3 + \frac{243}{8}h_x^4 + \frac{1671}{8}h_x^5 + \frac{22275}{16}h_x^6 + \frac{162855}{16}h_x^7 + \frac{9700617}{128}h_x^8 \\
& + \frac{595490847}{1024}h_x^9 + \frac{9308111103}{2048}h_x^{10} - \frac{3}{2}h_x h_z^2 - 3h_x^2 h_z^2 - \frac{219}{16}h_x^3 h_z^2 - \frac{2217}{32}h_x^4 h_z^2 \\
& - \frac{70533}{128}h_x^5 h_z^2 - \frac{2203115}{512}h_x^6 h_z^2 - \frac{442391465}{12288}h_x^7 h_z^2 - \frac{22361908135}{73728}h_x^8 h_z^2 - \frac{51}{8}h_x h_z^4 \\
& - \frac{423}{32}h_x^2 h_z^4 - \frac{11721}{128}h_x^3 h_z^4 - \frac{96375}{256}h_x^4 h_z^4 - \frac{2609429}{1024}h_x^5 h_z^4 - \frac{1202512427}{73728}h_x^6 h_z^4 + \frac{63}{4}h_z^6 \\
& - \frac{345}{8}h_x h_z^6 - \frac{23739}{512}h_x^2 h_z^6 - \frac{1561357}{2048}h_x^3 h_z^6 - \frac{305734169}{73728}h_x^4 h_z^6 + \frac{153}{2}h_z^8 - \frac{64425}{128}h_x h_z^8 \\
& - \frac{12120847}{24576}h_x^2 h_z^8 + \frac{53511}{32}h_z^{10}.
\end{aligned}
\tag{39}
$$

# B   DLogPadé extrapolants of low-field gap series

This appendix contains tables listing the critical point and the critical exponent $z\nu$ extracted from different DLogPadé extrapolants of the low-field gap series for different single parallel field directions. Tab. 1 yields information for $\vec{h} = (0,0,h_z)$, Tab. 2 yields information for $\vec{h} = (h_x,0,0)$ with $h_x > 0$, and Tab. 3 yields information for $\vec{h} = (h_x,0,0)$ with $h_x > 0$.

| L\M | 1 | 2 | 3 | 4 | 5 | 6 | 7 | 8 |
|---|---|---|---|---|---|---|---|---|
| 1 | 0.10752688 | 0.10547699 | 0.10523805 | 0.10509224 | 0.10504679 | 0.10500772 | 0.1049713 | 0.10495553 |
| 2 | 0.10472973 | 0.10523108 | 0.1045452 | 0.10503576 | 0.1055013 | 0.10491465 | 0.10491441 | |
| 3 | 0.1053412 | 0.10509749 | 0.10503222 | 0.10497918 | 0.10489671 | 0.10491441 | | |
| 4 | 0.10493661 | 0.10504401 | 0.10441976 | 0.10488783 | 0.10491162 | | | |
| 5 | 0.10508288 | 0.10501009 | 0.10489493 | 0.10492223 | | | | |
| 6 | 0.10493808 | 0.1049691 | 0.10491607 | | | | | |
| 7 | 0.10497757 | 0.10495585 | | | | | | |
| 8 | 0.10492934 | | | | | | | |

(a)

| L\M | 1 | 2 | 3 | 4 | 5 | 6 | 7 | 8 |
|---|---|---|---|---|---|---|---|---|
| 1 | 0.69372182 | 0.65821549 | 0.65298094 | 0.6489826 | 0.64748214 | 0.64597598 | 0.64434926 | 0.643553614 |
| 2 | 0.64097951 | 0.65278152 | 0.62099623 | 0.64702799 | 0.63621634 | 0.64078947 | 0.640772037 | |
| 3 | 0.65608076 | 0.649125 | 0.64687611 | 0.64453755 | 0.63942269 | 0.64077208 | | |
| 4 | 0.64357776 | 0.64736425 | 0.5775624 | 0.63870883 | 0.64055052 | | | |
| 5 | 0.64897912 | 0.646061499 | 0.63928539 | 0.64139252 | | | | |
| 6 | 0.64274506 | 0.64422605 | 0.64090402 | | | | | |
| 7 | 0.64468254 | 0.643556558 | | | | | | |
| 8 | 0.64202164 | | | | | | | |

(b)

Table 2: Results from the DLogPadé extrapolation of the flux gap at $\vec{k} = (0,0)$ for a single parallel field $\vec{h} = (0,0,h_x)$ with $h_x > 0$. (a) displays the critical field strength $h_{x,\mathrm{c}}$ while (b) shows the corresponding critical exponent.

| L\M | 1 | 2 | 3 | 4 | 5 | 6 | 7 | 8 |
|---|---|---|---|---|---|---|---|---|
| 1 | * | 0.33333333 | 0.30312643 | 0.31005218 | 0.30233198 | 0.31285739 | 0.2944598 | * |
| 2 | * | 0.29563016 | 0.30863659 | 0.30620068 | 0.30648033 | 0.30529545 | 0.30572772 | |
| 3 | * | 0.31505976 | 0.30536853 | 0.30646905 | 0.30624098 | 0.30562624 | | |
| 4 | * | 0.29660536 | 0.3067845 | 0.30583112 | 0.30553719 | | | |
| 5 | * | 0.32009256 | 0.30451723 | 0.30557599 | | | | |
| 6 | * | 0.27812109 | 0.30627714 | | | | | |
| 7 | * | 0.38512218 | | | | | | |
| 8 | * | | | | | | | |

(a)

| L\M | 1 | 2 | 3 | 4 | 5 | 6 | 7 | 8 |
|---|---|---|---|---|---|---|---|---|
| 1 | * | 0.88888889 | 0.70539914 | 0.7597267 | 0.68373644 | 0.82266529 | 0.57664826 | * |
| 2 | * | 0.6476255 | 0.74547404 | 0.72483014 | 0.72757445 | 0.71374833 | 0.71962492 | |
| 3 | * | 0.81342571 | 0.71623331 | 0.72743868 | 0.725145897 | 0.7180595 | | |
| 4 | * | 0.61761575 | 0.73136813 | 0.7204745 | 0.71669371 | | | |
| 5 | * | 0.94291929 | 0.70292324 | 0.7172923 | | | | |
| 6 | * | 0.37495053 | 0.72853634 | | | | | |
| 7 | * | 4.33852875 | | | | | | |
| 8 | * | | | | | | | |

(b)

Table 3: Results from the DLogPadé extrapolation of the flux gap at $\vec{k} = \pm(2\pi/3, -2\pi/3)$ for a single parallel field $\vec{h} = (0,0,h_x)$ with $h_x < 0$. (a) displays the critical field strength $h_{x,\mathrm{c}}$ while (b) shows the corresponding critical exponent. Asterisk mark values for [L,M], where the extrapolations failed due to complex poles.

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
