# Peer review of "Quantum robustness of the toric code in a parallel field on the honeycomb and triangular lattice"

_SciPost Physics_

## Round 2 · Referee Report · Anonymous (Referee 1) · 2024-4-8

Report
This is a nicely structured and well-written paper that presents a study of the honeycomb (equivalently, triangular) lattice toric code model in presence of longitudinal (x,z) fields. For completeness, the paper reviews several known results. The methodology is however relatively novel in this setting (SSE QMC, high-field series expansion and perturbative linked-cluster expansion with full graph decomposition), as are some of the findings -- including a thorough analysis of the scaling exponents, and the uncovering of an intriguing first order transition from the clock-ordered phase to the polarised phase for negative x-field values. The potential for multicritical behaviour is also intriguing.
Overall, I would like to recommend this manuscript for publication in SciPost Physics. It certainly meets all general acceptance criteria. I am asked to justify at least one of the 4 available expectations. I feel that "groundbreaking discovery" and "breathrough on a previous research stumbling bloch" do not apply, nor does really the "synergetic link between different areas". I am happy to make the case for a "new pathway in an existing research direction, with potential for multipronged follow-up", on the grounds of the comprehensive study and the application of techniques not commonly used in this research setting, and the discovery of a first order transition and possible multicritical behaviour. It could lead to interesting follow up work, but the case is admittedly not very strong in my opinion. No matter, I remain happy to recommend publication.
I have a couple of major points that I would like the authors to take into consideration (and a few possible typos spotted on the way):
1) in the abstract and in the manuscript, the authors claim that they "... demonstrate that all findings for the toric code on the honeycomb lattice can be transferred exactly to the toric code on the triangular lattice". I find this statement a bit strong and I would invite the authors to state it as an observation / fact rather than a demonstration. While it is perhaps true that no one has written it down on published paper before (I am not sure), it follows from a well-known and trivial property: the medial (i.e., bond-centred dual) lattice of the honeycomb and triangular lattice is the same (namely, the kagome lattice). If I am not mistaken, this is the key point in Sec.2.2, except then wishing to swap x and z spin components, which requires an additional rotation in spin space, resulting in the relabelling of the x and z components of the applied field (and putting a minus sign to the y component). Such mappings are extensively used in stat mech (thinking of Baxter's work on soluble models, off the top of my head, several decades ago).
2) everywhere the authors are careful with direct and dual lattice / strings / loops, except in between Eq.(4) and (5), where they talk about "any contractible loop of sigma^x or sigma^z matrices". It may be worth clarifying here also that in the latter case one ought to consider loops of the direct lattice (i.e., closed paths along the bonds) whereas in the former case one ought to consider loops of the dual lattice (i.e., closed paths that cross each bond of the direct lattice in its middle point, normal to it).
3) in Sec.3.1 the notation T_0, T_{\pm 2} is used without being properly introduced. It would be good to add a few lines of explanation / clarification for the unfamiliar reader.
4) in Fig.4, why is the extrapolated value visibly not on the linear fit line? Shouldn't it be the L-->infinity limit of said line?
5) In Sec.4, the authors find a direct first order transition between the clock-ordered phase and the polarised phase by investigating the behaviour of the charge-free and charge-full sectors only. The authors claim that, intuitively, other charge sectors do not alter the ground-state phase diagram. This is not immediately intuitive to me, and I was wondering if they could give a few further words of explanation. Given the curvature of the Pade result in Fig.6. One could imagine energies of finite charge density phases to also have positive curvature but sufficiently low overall value to intersect the blue curve to the left of the current transition and the red curve to the right of the current transition, thus inducing an intermediate phase with finite charge density. As a matter of fact, one could intriguingly speculate that there could be a staircase of phases of different finite-charge-density phases in between. Would it be much work to try, for example, the half-charge-filling case and see where the corresponding energy curve lies in Fig.6?
Possible typos:
Eq.(23) q_m in the denominator -- the m should be capitalised?
"To ensure, that" -- remove the comma?
"approximmants" -- typo
"quantum cluster update cluster in space" -- possible typo?
"descibed" -- typo
"physics of both, the low-field" -- I think that the comma is a typo
"phase transition is shift to large negative" -- typo, maybe "shifts"?

---

## Round 2 · Referee Report · Anonymous (Referee 2) · 2024-5-8

Strengths
(2) This is an extremely solid and well written work. The calculations are extensive and very well explained.
Weaknesses
Report
The paper is extremely solid work. The detailed duality maps and numerical calculations are well explained and written in a self-contained way.
I strongly recommend the publication of this work with no additional requested changes at this end. Inasmuch as the general listed expectations, this work ~ "opens a new pathway in an existing or a new research direction, with clear potential for multi-pronged follow-up work" and satisfies all of the required acceptance criteria.
Apart from the earlier comments, I had one other remark.
What is meant by "supersymmetry"? At this end, the symmetry that appears to be referred to under this name (that associated with the self-duality of the square lattice) does not immediately translate to conventional supersymmetry. If the intention of the authors was to implicitly discuss holes doped into the toric code model (which may indeed be supersymmetric, https://arxiv.org/pdf/1210.3232) then this might be made explicit.
Recommendation
Publish (easily meets expectations and criteria for this Journal; among top 50%)

---

## Round 3 · Author Response

Response to the "Anonymous Report 1 on 2024-4-8 (Invited Report)":

We thank the referee for the careful examination of our manuscript and the positive and constructive feedback.

1.) We thank the referee for this comment. In the revised version we have rephrased the corresponding sentence in the abstract as well as in Sect. 2.2.

2.) We thank the referee for this comment. It indeed is helpful to clarify, that the contractible loops of $sigma^x$ matrices lie on the direct lattice, while contractible loops of $sigma^z$ matrices lie on the dual lattices. In the revised version we specified, on which lattice the loops lie.

3.) We thank the referee for this comment. We have now introduced the $T$-operators in a way that it is easier to grasp for unfamiliar readers.

4.) We thank the referee for this comment. Indeed, the data depicted in the plot was wrong, due to erroneous plotting. We corrected our error and replaced the figure with the correct one.

5.) We thank the referee for this comment. In the revised version we have removed the sentence regarding the intuition that only the charge full and empty sector are relevant. We have further stressed that we only focused on these two sectors in the article. Let us mention that we started working on a quantum Monte Carlo technique that is directly sampling the toric code in a $XZ$-field to explore the role of fractional charge fillings.

We thank the referee for these issues and the list of typos.

We have corrected all suggested typos in the revised version.

Response to the "Anonymous Report 2 on 2024-5-8 (Invited Report) ":

We thank the referee for the careful examination of our manuscript and the positive feedback.

We thank the referee for his comment. We used the term supersymmetry to refer to the symmetry of the toric code on a square lattice when interchanging x and z fields (as done frequently in the literature). In the revised version we have dropped the notion "supersymmetry" and replaced it with "symmetry" to avoid misunderstandings.

---

## Round 3 · List of Changes

Line 21-22: Changed "We further demonstrate that all findings for the toric code on the honeycomb lattice can be transferred exactly to the toric code on a triangular lattice." to "All findings for the toric code on the honeycomb lattice can be transferred exactly to the toric code on the triangular lattice."

Line 75: Changed "supersymmetry" to "symmetry"

Line 78: Changed "supersymmetry" to "symmetry"

Line 94: Changed "demonstrate" to "observe"

Line 126: Changed "Furthermore, any contractible loop of $\sigma^x$ or $\sigma^z$ matrices corresponds to the product of operators X or Z contained in the loop, respectively [15]." to "Furthermore, any contractible loop of $\sigma^z$ matrices on the honeycomb lattice or $\sigma^x$ matrices on its dual triangular lattice corresponds to the product of operators X or Z contained in the loop, respectively [15]."

Line 167: Changed "First, observe that the triangular lattice is dual to the honeycomb lattice, and that the very same spin sites reside on the links of both lattices." to "Indeed, the triangular lattice is dual to the honeycomb lattice and the very same spin sites reside on the links of both lattices."

Line 194: Changed "In the low-field expansion, we can express the parallel field as $T_0 + T_{+2} + T_{-2}$ , where the commutation relation $[H_{\rm tc} , T_n ] = nT_n$ holds true, with $n$ representing the net change in total charge and flux particle numbers resulting from the action of $T_n$" to "In the low-field expansion, we can express the field term $\sum_i h_x \sigma_i^x$ as $T_0^{\rm f} + T_{+2}^{\rm f} + T_{-2}^{\rm f}$, since a field in $x$-direction either creates a flux-pair $T_{+2}$, annihilates a flux-pair $T_{-2}$ or moves a single flux by $T_0$ on the respective bonds. Similarly, the field in $z$-direction $\sum_i h_z \sigma_i^z$ can be expressed as $T_0^{\rm c} + T_{+2}^{\rm c} + T_{-2}^{\rm c}$ , as it creates a charge-pair, annihilates a charge-pair or moves a charge on the respective bonds. In both cases, $T_n = T_n^{\rm f} +T_n^{\rm c}$ fullfills the commutation relation $[H_{\rm tc} , Tn ] = nT_n$, with $n$ representing the net change in total charge and flux particle numbers resulting from the action of $T_n$."

Line 502: Added "At this point it should be stressed, that we only considered the charge-free and charge-full sectors, while all other sectors of fractional filling were neglected. A more thorough analysis is needed, since the tools presented in this work do not allow for an examination of those sectors."

We updated Figure 4

---

## Editorial Decision

accepted_in_target_journal